# Neoadjuvant Pembrolizumab Plus Chemotherapy in Early-Stage Triple-Negative Breast Cancer: A Nationwide Retrospective Turkish Oncology Group Study

**DOI:** 10.3390/cancers16193389

**Published:** 2024-10-03

**Authors:** Ebru Karci, Ahmet Bilici, Buket Bayram, Melisa Celayir, Neslihan Ozyurt, Başak Oyan Uluc, Aynur Eken, Gul Basaran, Umut Demirci, Yasemin Kemal, Mehmet Berk Oruncu, Omer Fatih Olmez, Fatih Selcukbiricik, Taner Korkmaz, Ismail Erturk, Irem Bilgetekin, Serkan Celik, Alper Turkel, Ali Alkan, Abdullah Sakin, Orcun Can, Meral Gunaldi, Ece Esin, Ozcan Yildiz

**Affiliations:** 1Department of Medical Oncology, Faculty of Medicine, İstanbul Medipol University, Istanbul 34214, Türkiye; abilici@medipol.edu.tr (A.B.); omerfatih.olmez@medipol.com.tr (O.F.O.); abdullah.sakin@medipol.edu.tr (A.S.); oyildiz@medipol.edu.tr (O.Y.); 2Department of Medical Oncology, Koc University Hospital, Istanbul 34010, Türkiye; bkuvvet@kuh.ku.edu.tr (B.B.); fselcukbiricik@kuh.ku.edu.tr (F.S.); 3Department of Medical Oncology, Faculty of Medicine, Mehmet Ali Aydınlar Acıbadem University, Istanbul 34752, Türkiye; ozde.celayir@acibadem.com (M.C.); basak.uluc@acibadem.com (B.O.U.); gul.basaran@acibadem.edu.tr (G.B.); taner.korkmaz@acibadem.com (T.K.); orcun.can@acibadem.com (O.C.); 4Department of Medical Oncology, Faculty of Medicine, Ordu University Training and Research Hospital, Ordu 52200, Türkiye; neslihanozyurt@odu.edu.tr; 5Department of Medical Oncology, Ozel Ortadogu Hospital, Adana 67055, Türkiye; ekenaynur@gmail.com; 6Medical Oncology Unit, Memorial Ankara Hospital, Ankara 06520, Türkiye; umut.demirci@memorial.com.tr (U.D.); irem.bilgetekin@memorial.com.tr (I.B.); 7Department of Medical Oncology, Faculty of Medicine, Altınbas University, Istanbul 34147, Türkiye; yasemin.kemal@medikalpark.com.tr; 8Department of Medical Oncology, Faculty of Medicine, Ankara University, Ankara 06100, Türkiye; mboruncu@ankara.edu.tr; 9Ministry of Health Gülhane Training and Research Hospital, Ankara 06010, Türkiye; ismail.erturk@sbu.edu.tr; 10Department of Medical Oncology, Yeditepe University, Istanbul 34755, Türkiye; celik.serkan@yeditepe.edu.tr; 11Department of Medical Oncology, Abdurrahman Yurtaslan Ankara Oncology Research and Training Hospital, Ankara 06200, Türkiye; alper.turkel@sbu.edu.tr; 12Department of Medical Oncology, School of Medicine, Mugla Sıtkı Kocman University, Mugla 48000, Türkiye; alialkan@mu.edu.tr; 13Department of Medical Oncology, Faculty of Medicine, Aydın University, Istanbul 34295, Türkiye; meral.gunaldi@medicana.com.tr; 14Department of Medical Oncology, Bayındır Hospital, Ankara 06250, Türkiye; dr.eceesin@gmail.com

**Keywords:** triple-negative breast cancer, neoadjuvant therapy, pembrolizumab, pathological complete response, real world, event-free survival, overall survival, safety

## Abstract

**Simple Summary:**

This study investigates the real-world efficacy and safety of combining pembrolizumab, a novel immunotherapy agent, with chemotherapy in early-stage triple-negative breast cancer treatment. We specifically aimed to validate clinical trial results in routine practice. A total of 108 Turkish patients receiving neoadjuvant therapy were examined. The combined regimen demonstrated high efficacy, with 64% of patients achieving pathological complete response, and exhibited generally favorable safety profiles with predominantly mild adverse events. These findings support the use of this combination as a standard treatment for this aggressive breast cancer subtype. However, the results underscore the need for further research to identify optimal patient selection criteria, which can inform oncologists’ decision-making and potentially enhance outcomes for patients with triple-negative breast cancer.

**Abstract:**

**Background/Objectives:** Following the results of the phase 3 KEYNOTE-522 trial, the U.S. Food and Drug Administration approved pembrolizumab, a humanized IgG4 kappa monoclonal antibody, in combination with neoadjuvant chemotherapy as a new standard of care for high-risk early-stage triple-negative breast cancer (TNBC). This retrospective, multicenter study in Türkiye assessed the real-world efficacy and safety of neoadjuvant pembrolizumab combined with chemotherapy in early-stage TNBC. **Methods:** The study included 108 patients treated between 2021 and 2023 across 14 oncology centers. Three distinct neoadjuvant regimens incorporating pembrolizumab were administered at the discretion of the treating physicians. The primary outcomes were the pathological complete response (pCR) rate after neoadjuvant therapy and the 2-year event-free survival (EFS) and overall survival (OS) rates. **Results:** The observed pCR rate was 63.9%, closely mirroring the 64.8% reported in the KEYNOTE-522 trial. At the two-year mark, the EFS rate was 87.2% and the OS rate was 92.3%. Multivariable analysis identified pCR as the sole independent predictor of both EFS and OS. The safety profile was consistent with previous clinical trial data, with most adverse events being of grade 1–2 in severity. **Conclusions:** These findings provide valuable real-world confirmation of the efficacy and safety of neoadjuvant pembrolizumab–chemotherapy in early-stage TNBC, complementing evidence from randomized trials.

## 1. Introduction

Triple-negative breast cancer (TNBC) constitutes approximately 10–15% of breast cancer diagnoses and is characterized by the absence of estrogen receptor, progesterone receptor, and human epidermal growth factor receptor 2 expression [1,2]. The treatment of TNBC remains challenging, even in early stages, due to its biological aggressiveness and the limited availability of targeted therapies, resulting in higher rates of relapse and increased mortality risk [3,4,5]. Consequently, the primary objective of systemic therapy in non-metastatic TNBC is to mitigate the risk of distant recurrence and death [6]. Historically, the principal advantage of neoadjuvant therapy in TNBC was the improvement of surgical outcomes through tumor downstaging in both the breast and axillary lymph nodes [7]. Thus, it may avoid unnecessary axillary lymph node dissection (AD) and make breast-conserving surgery (BCS) possible for patients who were initially considered for mastectomy. In the preoperative setting, BCS has been proven to be a safe alternative to mastectomy in terms of long-term oncologic outcomes and survival [8].

However, there is currently a strong consensus favoring neoadjuvant therapy as the preferred treatment approach for early-stage TNBC [9]. Crucially, patients achieving a pathological complete response (pCR) post-neoadjuvant therapy exhibit significantly better survival outcomes, including extended event-free survival (EFS) and overall survival (OS) [10]. As a result, regulatory agencies presently endorse the use of pCR as a surrogate endpoint in clinical trials evaluating neoadjuvant therapies for early-stage breast cancer [11].

In 2021, based on the pivotal phase 3 KEYNOTE-522 trial results [12], the U.S. Food and Drug Administration approved pembrolizumab, a humanized IgG4 kappa monoclonal antibody, in combination with neoadjuvant chemotherapy as a new standard of care for high-risk, early-stage TNBC. In the KEYNOTE-522 trial, the results demonstrated benefits in both neoadjuvant and adjuvant settings [11]. The pCR rate was significantly higher in the pembrolizumab–chemotherapy group (64.8%) compared to the placebo–chemotherapy group (51.2%), reflecting the neoadjuvant benefit [11]. After a median follow-up of 39.1 months, the estimated 36-month EFS was 84.5% in the pembrolizumab–chemotherapy group versus 76.8% in the placebo–chemotherapy group [13]. This EFS benefit, resulting from the entire treatment course (neoadjuvant and adjuvant), was consistent across various subgroups, including those defined by PD-L1 expression and nodal involvement [13].

This nationwide, multicenter retrospective study conducted in Türkiye sought to evaluate the real-world clinical benefits of neoadjuvant pembrolizumab combined with chemotherapy in early-stage TNBC. Specifically, we assessed pCR rates to evaluate the efficacy of neoadjuvant regimens while investigating survival outcomes, including EFS and OS, to examine the efficacy of the entire treatment course. We also performed analyses to identify potential markers for treatment response and evaluate the safety profile of the implemented treatment protocols within this clinical population.

## 2. Methods

### 2.1. Study Design

This retrospective study, based on a review of clinical charts, was conducted between 2021 and 2023 across 14 oncology centers in Türkiye. Adult patients aged > 18 years were eligible if they met the following criteria: (1) newly diagnosed with previously untreated, non-metastatic TNBC with tumor staging of T1c (N1–2) or T2–4 (N0–2) according to the American Joint Committee on Cancer staging manual (seventh edition), (2) centrally confirmed TNBC in all foci as defined by the guidelines of the American Society of Clinical Oncology and the College of American Pathologists, and (3) an Eastern Cooperative Oncology Group (ECOG) performance status [14] score of 0 or 1. Patients with multifocal primary tumors and inflammatory breast cancers were also eligible. Notably, patient eligibility was independent of PD-L1 status. The exclusion criteria were as follows: (1) active autoimmune diseases requiring systemic treatment within the previous two years, (2) a diagnosis of immunodeficiency, (3) use of immunosuppressive therapy within the week prior to enrollment, (4) a history of human immunodeficiency virus infection, (5) non-infectious pneumonitis treated with glucocorticoids, (6) current pneumonitis, (7) any active infection, and (8) clinically significant cardiovascular disease. The final study cohort comprised 108 patients. Ethical approval was granted by the Institutional Review Board at Medipol Mega University Hospital (Istanbul, Türkiye) under the reference number 10840098-202.3.02-2005. Written informed consent was obtained from all patients or their designated legal representatives.

### 2.2. Neoadjuvant Treatment

In the neoadjuvant phase, three distinct regimens incorporating pembrolizumab were administered at the discretion of the treating physician. The first regimen consisted of pembrolizumab (200 mg) given intravenously every 3 weeks, in combination with doxorubicin (60 mg/m^2^) and cyclophosphamide (600 mg/m^2^) administered every 2 weeks for four cycles. This was followed by four additional cycles of pembrolizumab every 3 weeks, combined with weekly paclitaxel (80 mg/m^2^) and carboplatin (area under the curve (AUC) = 1.5 mg/mL/min) for 12 weeks. The second regimen involved pembrolizumab (200 mg) every 3 weeks, alongside doxorubicin (60 mg/m^2^) and cyclophosphamide (600 mg/m^2^) every 2 weeks for four cycles. This was followed by four cycles of pembrolizumab combined with paclitaxel (175 mg/m^2^) and carboplatin (AUC = 5 mg/mL/min) every 3 weeks. The third regimen was similar to the first but with adjusted timing. Patients received pembrolizumab (200 mg), doxorubicin (60 mg/m^2^), and cyclophosphamide (600 mg/m^2^) every 3 weeks for four cycles. This was followed by four cycles of pembrolizumab every 3 weeks, combined with weekly paclitaxel (80 mg/m^2^) and carboplatin (AUC = 1.5 mg/mL/min) for 12 weeks.

### 2.3. Definitive Surgery and Adjuvant Treatment

Definitive surgery, consisting of either breast conservation or mastectomy with sentinel lymph node evaluation or axillary dissection, was performed 3 to 6 weeks after the completion of the neoadjuvant phase. In the adjuvant setting, patients were offered radiation therapy as clinically indicated, along with pembrolizumab treatment administered every 3 weeks for up to nine cycles, based on the physician’s discretion. Patients with residual postoperative disease were also offered adjuvant capecitabine treatment. Treatment discontinuation occurred in cases of disease progression, recurrence, or unacceptable toxicity.

### 2.4. Data Collection

Demographic data, including age and menopausal status, were recorded for each patient. Performance status was evaluated using the ECOG scale [14]. Tumor-related information, such as primary lesion characteristics, histopathological type, Ki-67 index, PD-L1 status [15], tumor stage, lymph node status, and overall disease stage, was collected. Treatment-related data encompassed the schedule of carboplatin administration, the type of surgery performed, and the use of adjuvant pembrolizumab and capecitabine. *BRCA*1/2 mutation status was available for a subset of patients.

### 2.5. Efficacy of Neoadjuvant Regimens

Response to neoadjuvant treatment was evaluated using the Response Evaluation Criteria in Solid Tumors (RECIST) guidelines, version 1.1 [16]. The primary endpoint for evaluating the efficacy of neoadjuvant therapy was pCR, defined as no evidence of primary tumor or in situ carcinoma and no regional lymph node involvement (ypT0/Tis, N0) [17].

### 2.6. Survival Outcomes

To evaluate the efficacy of the entire treatment course, EFS and OS were analyzed. EFS was defined as the time interval from the initiation of treatment to the occurrence of any of the following events: disease progression precluding definitive surgery, local or distant recurrence, diagnosis of a second primary malignancy, or death from any cause. OS was measured as the time from treatment initiation until the patient’s death, regardless of the underlying cause. For patients who remained alive at the time of analysis, OS was censored at the date of the last follow-up assessment.

### 2.7. Safety Assessment

The severity and incidence of treatment-related adverse events (TRAEs) were evaluated using version 5.0 of the National Cancer Institute Common Terminology Criteria for Adverse Events (NCI-CTCAE) [18]. The incidence of TRAEs was reported separately for grade 1–2 (mild to moderate) and grade 3–4 (severe to life-threatening) events. For this analysis, specific TRAEs of interest were identified, including gastrointestinal symptoms such as nausea, vomiting, diarrhea, stomatitis, and constipation; neurological effects like peripheral neuropathy and fatigue; metabolic disturbances such as decreased appetite, adrenal insufficiency, and hypothyroidism; hematological abnormalities, including thrombocytopenia, anemia, and neutropenia; hepatic enzyme elevations; and dermatological manifestations like rash and alopecia. The percentage of patients experiencing each specific TRAE was recorded for the entire study cohort.

### 2.8. Data Analysis

The data collected from all participating centers were aggregated for analysis. Descriptive statistics, including counts, percentages, means, standard deviations, medians, and ranges, were employed to express the variables. Kaplan–Meier plots were generated to depict survival estimates, and statistical comparisons were performed using the log-rank test. Multivariable logistic regression analysis was employed to identify independent predictors of achieving a pCR following neoadjuvant treatment. To investigate the associations between the variables under examination and survival outcomes, both univariable and multivariable Cox proportional hazards regression analyses were performed. A stepwise selection approach was implemented, incorporating variables that demonstrated significance in the univariable analysis into the multivariable model. The results are presented as hazard ratios (HRs) accompanied by their 95% confidence intervals (CIs). The SPSS software, version 24.0 (IBM, Armonk, NY, USA), was employed to conduct the analyses. Statistical significance was defined by two-tailed *p* values below 0.05.

## 3. Results

### 3.1. Patient Characteristics

The general characteristics of the 108 patients with early-stage TNBC are detailed in Table 1. The majority (91.7%) of patients were under 65 years old, with 60.2% being premenopausal. Most patients (81.5%) had an ECOG performance status of 0. Regarding disease characteristics, 75% of patients had single primary lesions, and the predominant histopathological type was invasive ductal carcinoma (82.4%). A high proportion (93.5%) of patients exhibited a Ki-67 index greater than 20%. The majority of patients (82.4%) presented with T1–T2 stage primary tumors and 65.7% had positive lymph node status. Stage II disease was the most common, accounting for 58.3% of cases. Regarding neoadjuvant carboplatin administration, 69.4% of patients received the treatment on a weekly basis, while the remaining 30.6% adhered to a triweekly regimen. The most frequent type of surgery was breast-conserving surgery (BCS) with sentinel lymph node biopsy (SLNB) (50.9%). BCS was performed in 29 (26.8%) patients who intended to undergo modified radical mastectomy (MRM) at the time of diagnosis after neoadjuvant treatment. On the other hand, SNLB was performed after neoadjuvant treatment in 51 of 71 patients (71.8%) with clinical/histological lymph node positivity. All of the remaining 35 patients who underwent SLNB were clinically lymph node-negative. Of the 22 patients who underwent axillary dissection (AD), 5 (22.7%) had AD without SLNB. The remaining 17 patients had AD after SLNB. Only four of the patients who underwent SLNB had positive N1a lymph nodes. Adjuvant therapy data indicated that 30.6% of patients received adjuvant pembrolizumab, while 35.2% received adjuvant capecitabine. Regarding molecular characteristics, 39.8% of patients were PD-L1-positive and 13% harbored *BRCA*1/2 mutations. However, *BRCA*1/2 mutation status was undetermined for a significant proportion (39.8%) of the cohort.

### 3.2. Response to Neoadjuvant Treatment

The majority of patients (61.1%, n = 66) underwent the first neoadjuvant regimen, whereas the second scheme was administered to a small subset of five patients (4.6%). The third regimen was assigned to 37 patients, accounting for 34.3% of the study population. A substantial proportion of patients (63.0%, n = 68) achieved a complete clinical response after neoadjuvant therapy. Partial clinical response was observed in 28.7% (n = 31) of patients, while 8.3% (n = 9) experienced stable disease. Notably, no patients exhibited progressive disease following the neoadjuvant treatment period. The primary efficacy endpoint, pCR, was achieved in 63.9% (n = 69) of patients, whereas 36.1% (n = 39) did not attain pCR. No differences were found in pCR rates among the three neoadjuvant chemotherapy regimens (*p* = 0.32).

### 3.3. Survival Outcomes

The median follow-up duration for the 108 participants was 19.8 months (range: 8−28 months). At the two-year mark, the EFS rate among the study patients was 87.2%, whereas the OS rate was 92.3%. Table 2 presents the results of univariable and multivariable Cox regression analyses for predicting 2-year EFS. Univariable analyses identified significant associations with ECOG performance status, carboplatin scheduling, *BRCA*1/2 mutation status, adjuvant capecitabine, and the achievement of pCR following neoadjuvant therapy. After adjusting for potential confounders, multivariable analysis indicated that only pCR retained an independent association with 2-year EFS, with an HR of 5.90 (95% CI = 1.17–9.87; Figure 1). Table 3 details the univariable and multivariable Cox regression analyses for predicting 2-year OS. The univariable predictors of 2-year OS were consistent with those identified for 2-year EFS. In the multivariable model, pCR emerged as the only significant predictor of 2-year OS, with an HR of 1.62 (95% CI = 1.02–2.64; Figure 2).

### 3.4. Safety

Table 4 summarizes the most commonly reported TRAEs during treatment. The majority were of grade 1 or 2 in severity. The most frequent grade 1–2 events included alopecia (69.4%), nausea (64.8%), neutropenia (48.1%), fatigue (41.6%), vomiting (33.3%), and anemia (33.3%). Grade 3 or 4 adverse events were less common, with the most prevalent being neutropenia (16.6%), anemia (6.4%), and fatigue, decreased appetite, increased ALT, and nausea (all at 3.7%). Notably, no grade 3–4 events were reported for stomatitis, thrombocytopenia, rash, or constipation. Some less frequent but noteworthy adverse events included hypothyroidism (11.1%, grade 1–2; 1.8%, grade 3–4) and adrenal insufficiency (0.9%, grade 1–2).

## 4. Discussion

The integration of pembrolizumab with chemotherapy in a neoadjuvant setting has emerged as a significant advancement in the treatment of early-stage TNBC [19]. This retrospective, multicenter study conducted in Türkiye provides valuable real-world evidence on the efficacy and safety of this combination therapy, complementing data from controlled clinical trials [12,13]. The investigation yielded several key findings that contribute to our understanding of this therapeutic approach. First, the observed pCR rate of 63.9% following neoadjuvant therapy corroborates the efficacy of incorporating pembrolizumab into neoadjuvant regimens for patients with early-stage TNBC in real-world clinical settings. Second, the inability to identify independent predictors of pCR within the cohort underscores the inherent complexity in stratifying patients most likely to benefit from neoadjuvant pembrolizumab in routine practice. Third, the achievement of pCR following neoadjuvant therapy emerged as the sole independent predictor of both EFS and OS. Lastly, the safety profile observed in this study closely aligned with previously reported clinical trial data on pembrolizumab plus chemotherapy in TNBC [12,13]. This consistency reinforces the feasibility and tolerability of integrating immunotherapy into neoadjuvant regimens within standard clinical practice, providing reassurance to clinicians regarding the management of potential TRAEs.

The observed pCR rate of 63.9% in our study aligns closely with the 64.8% rate reported in the KEYNOTE-522 trial [11]. Furthermore, our findings in Türkiye are consistent with the subanalysis conducted by Takahashi et al. [20], which reported a pCR rate of 58.7% in patients with early-stage TNBC from East/Southeast Asia (Korea, Japan, Taiwan, and Singapore). A noteworthy observation in our real-world study is the utilization of three distinct neoadjuvant regimens incorporating pembrolizumab, each differing in dosing schedules, drug combinations, and timing of administration. This raises a question regarding the optimal chemotherapeutic regimen when using pembrolizumab in the neoadjuvant setting, particularly in efforts to mitigate toxicity [21]. For instance, the phase II NeoPACT trial, which evaluated 109 patients with TNBC receiving six cycles of neoadjuvant carboplatin and docetaxel with pembrolizumab, achieved a 58% pCR rate with an anthracycline-free regimen [22]. Our analysis did not identify a specific neoadjuvant regimen predictive of pCR. While this finding warrants cautious interpretation, it underscores the necessity for future research aimed at optimizing the balance between toxicity and efficacy of preoperative chemotherapy regimens in combination with pembrolizumab. Notably, our study did not reveal any independent predictors of pCR. This outcome contrasts with previous data suggesting that markers such as Ki-67 and PD-L1 expression may influence the response to neoadjuvant therapy in TNBC [23]. The absence of predictive factors in our cohort highlights the complex and heterogeneous nature of TNBC [24], emphasizing the need for further investigations to identify reliable biomarkers for treatment response. Recent studies have explored various potential biomarkers, including tumor-infiltrating lymphocytes and genomic signatures [23]; however, their clinical utility remains to be fully established.

A critical consideration that warrants further examination is the uncertainty surrounding the necessity of administering both neoadjuvant and adjuvant pembrolizumab to achieve long-term survival benefits [21]. Our investigation revealed that adjuvant pembrolizumab did not significantly impact EFS or OS. Conversely, the emergence of pCR following neoadjuvant treatment as the sole independent predictor of both survival outcomes in our study underscores its crucial long-term prognostic significance, a finding that is in accordance with the published literature [25,26]. Regarding safety considerations, the majority of adverse events observed in our cohort were of grade 1–2 severity and were determined to be manageable, supporting the tolerability profile of pembrolizumab in combination with chemotherapy in real-world clinical settings. The spectrum and severity of adverse events were generally consistent with those reported in the KEYNOTE-522 trial [12,13]. Notably, the incidence of immune-related adverse events, such as hypothyroidism and adrenal insufficiency, was relatively low, highlighting the feasibility of incorporating immunotherapy into neoadjuvant regimens for TNBC.

There are several limitations inherent to this study. The retrospective nature of the research design introduces potential biases and the influence of residual confounding factors, which may impact the reliability of the findings. To address this limitation, future prospective studies could provide more robust and compelling evidence to support the conclusions drawn.

A notable constraint of the study is its relatively modest sample size, particularly in terms of stage I patient representation. This limitation is especially pertinent given that the KEYNOTE-522 trial solely focused on stage II–III TNBC [12,13]. Consequently, there is a clear need for larger-scale studies specifically targeting stage I TNBC to fill this knowledge gap and provide more comprehensive insights. Another significant caveat is the absence of patient-reported outcomes (PROs). These measures are crucial in evaluating cancer treatments holistically, as they provide valuable insights into the patient experience beyond clinical outcomes [27]. Incorporating PROs in future research would offer a more nuanced and comprehensive assessment of the benefits and risks associated with pembrolizumab plus neoadjuvant chemotherapy. Finally, as a real-world data study, this investigation lacked a control arm, which is a common limitation in similar real-life studies [28]. The absence of a control group makes it challenging to definitively attribute observed outcomes to the treatment under investigation, as there is no direct comparison to standard care or alternative treatments.

## 5. Conclusions

Our findings provide valuable real-world confirmation of the efficacy and safety of neoadjuvant pembrolizumab–chemotherapy in early-stage TNBC, complementing evidence from randomized trials. This real-world study reinforces the role of pembrolizumab plus chemotherapy as a standard neoadjuvant treatment for early-stage TNBC, demonstrating high pCR rates that translate into improved survival outcomes. The absence of reliable predictive factors for pCR underscores the need for further research to enable more personalized treatment approaches. Ongoing trials comparing different combination strategies and biomarker-driven patient selection will be crucial to further optimize outcomes in this aggressive breast cancer subtype.

## Figures and Tables

**Figure 1 cancers-16-03389-f001:**
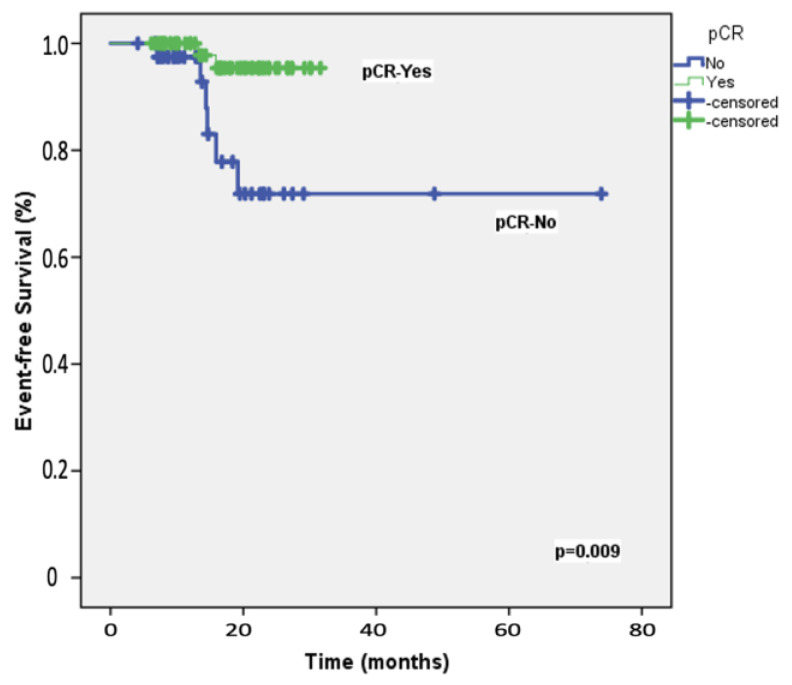
Event-free survival: Kaplan–Meier plot illustrating event-free survival in the 108 study patients, stratified by the achievement of pathologic complete response (pCR) following neoadjuvant therapy.

**Figure 2 cancers-16-03389-f002:**
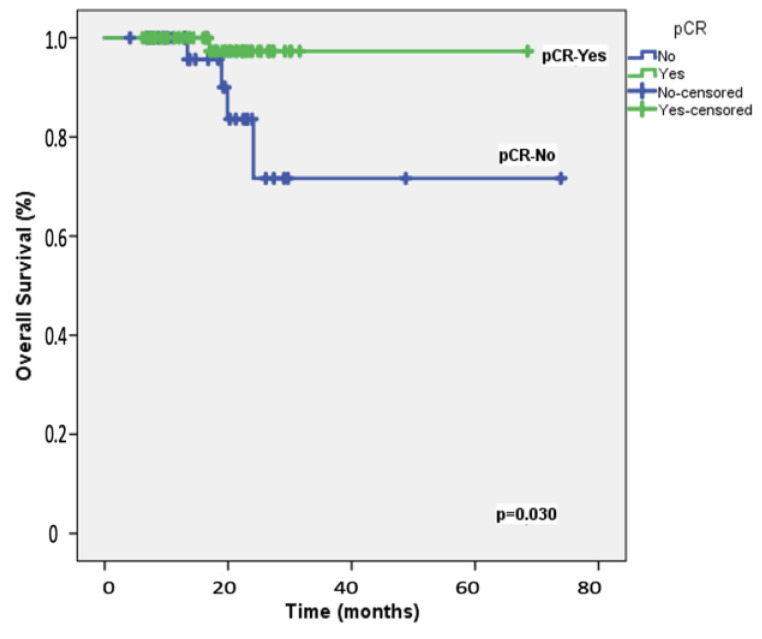
Overall survival: Kaplan–Meier plot depicting overall survival in the 108 study patients, stratified by the achievement of pathologic complete response (pCR) following neoadjuvant therapy.

**Table 1 cancers-16-03389-t001:** General characteristics of patients (n = 108) with early-stage triple-negative breast cancer.

Variable	n (%)
Age, years; median (range)	45 (27−85)
<65	99 (91.7)
≥65	9 (8.3)
Menopausal status	
Premenopausal	65 (60.2)
Postmenopausal	43 (39.8)
ECOG performance status	
0	88 (81.5)
1	20 (18.5)
Primary lesion	
Single	81 (75.0)
Multifocal	17 (15.7)
Multicentric	10 (9.3)
Histopathological type	
IDC	89 (82.4)
Invasive carcinoma, NOS	13 (12.1)
Metaplastic carcinoma	5 (4.6)
Other	1 (0.9)
Ki-67 index	
<20%	7 (6.5)
>20%	101 (93.5)
Primary tumor T stage	
T1–T2	89 (82.4)
T3–T4	19 (17.6)
Primary lymph nodal status	
Node-negative	37 (34.3)
Node-positive	71 (65.7)
Primary disease stage	
I	3 (2.8)
II	63 (58.3)
III	42 (38.9)
PD-L1 status	
Negative	65 (60.2)
Positive	43 (39.8)
Schedule of carboplatin	
Weekly	75 (69.4)
Every 3 weeks	33 (30.6)
Surgery type	
BCS-SLNB	55 (50.9)
BCS-AD	13 (12.1)
SCM-SLNB	31 (28.7)
MRM	9 (8.3)
BRCA1/2 mutation status	
Negative	51 (47.2)
Positive	14 (13.0)
Unknown	43 (39.8)
Adjuvant pembrolizumab	
No	75 (69.4)
Yes	33 (30.6)
Adjuvant capecitabine	
No	70 (64.8)
Yes	38 (35.2)

Data are expressed as counts with percentages in parentheses, unless otherwise indicated. Abbreviations: ECOG—Eastern Cooperative Oncology Group; IDC—invasive ductal carcinoma; NOS—not otherwise specified; PDL-1-Programmed Death Ligand Ligand-1; BCS-SLNB—breast-conserving surgery with sentinel lymph node biopsy; BCS-AD—breast-conserving surgery with axillary dissection; SCM-SLNB—skin-sparing mastectomy with sentinel lymph node biopsy; MRM—modified radical mastectomy.

**Table 2 cancers-16-03389-t002:** Univariable and multivariable analysis of event-free survival.

Variable	n (%)	2-Years EFS (%)	Univariable *p* Value	Multivariable *p* Value	HR (95% CI)
Age, years					
<65	99 (91.7)	89.7	**0.09**	0.96	1.76 (0.54–3.35)
≥65	9 (8.3)	59.3			
Menopausal status					
Premenopausal	65 (60.2)	85.7	0.53		
Postmenopausal	43 (39.8)	89.4			
ECOG performance status					
0	88 (81.5)	91.9	**0.021**	0.96	0.89 (0.51–3.54)
1	20 (18.5)	66.8			
Primary lesion					
Single	81 (75.0)	82.9	0.23		
Multifocal	17 (15.7)	84.8			
Multicentric	10 (9.3)	NA			
Histopathological type					
IDC	89 (82.4)	86.0	0.56		0.33 (0.13–0.85)
Invasive carcinoma, NOS	13 (12.1)	NA			
Metaplastic carcinoma	5 (4.6)	75.0			
Other	1 (0.9)	NA			
Primary tumor T stage					
T1–T2	89 (82.4)	84.2	0.15		2.0 (0.81–4.9)
T3–T4	19 (17.6)	NA			
Primary lymph node status					
Node-negative	37 (34.3)	94.4	0.26		
Node-positive	71 (65.7)	84.1			
Primary disease stage					
I	3 (2.8)	NA	0.99		
II	63 (58.3)	85.5			
III	42 (38.9)	89.6			
Schedule of carboplatin					
Weekly	75 (69.4)	91.1	**0.032**	0.28	1.11 (0.88–5.41)
Every 3 weeks	33 (30.6)	69.6			
PD-L1 status					
Negative	65 (60.2)	81.8	0.10		
Positive	43 (39.8)	95.8			
BRCA1/2 mutation status					
Negative	51 (47.2)	89.2	**0.043**	0.12	9.51 (0.54–27.1)
Positive	14 (13.0)	69.9			
Unknown	43 (39.8)	71.2			
Adjuvant pembrolizumab					
No	75 (69.4)	86.8	0.64		
Yes	33 (30.6)	86.6			
Adjuvant capecitabine					
No	70 (64.8)	75.1	**0.014**	0.90	1.28 (0.50–5.48)
Yes	38 (35.2)	97.3			
pCR					
Absent	39 (36.1)	71.9	**0.009**	**0.031**	5.90 (1.17–9.87)
Present	69 (63.9)	95.4			

Significant *p* values are highlighted in bold. Variables identified as significant in the univariable analysis were subsequently entered into the multivariable model. Abbreviations: ECOG—Eastern Cooperative Oncology Group; IDC—invasive ductal carcinoma; NOS—not otherwise specified; PD-L1—programmed death-ligand 1; *BRCA*1/2—breast cancer gene 1/2; *pCR*—pathological complete response; EFS—event-free survival; HR—hazard ratio; CI—confidence interval; NA—not available.

**Table 3 cancers-16-03389-t003:** Univariable and multivariable analysis of overall survival.

Variable	n (%)	2-Years OS (%)	Univariable *p* Value	Multivariable *p* Value	HR (95% CI)
Age, years					
<65	99 (91.7)	93.4	0.44		
≥65	9 (8.3)	85.7			
Menopausal status					
Premenopausal	65 (60.2)	91.4	0.54		
Postmenopausal	43 (39.8)	95.8			
ECOG performance status					
0	88 (81.5)	97.1	**0.021**	0.76	0.12 (0.05–2.54)
1	20 (18.5)	77.0			
Primary lesion					
Single	81 (75.0)	89.9	0.36		
Multifocal	17 (15.7)	NA			
Multicentric	10 (9.3)	NA			
Histopathological type					
IDC	89 (82.4)	93.6	0.53		
Invasive carcinoma, NOS	13 (12.1)	89.4			
Metaplastic carcinoma	5 (4.6)	NA			
Other	1 (0.9)	NA			
Primary tumor T stage					
T1–T2	89 (82.4)	90.9	0.24		
T3–T4	19 (17.6)	NA			
Primary lymph node status					
Node-negative	37 (34.3)	NA	0.12		
Node-positive	71 (65.7)	89.4			
Primary disease stage					
I	3 (2.8)	NA	0.68		
II	63 (58.3)	94.7			
III	42 (38.9)	84.4			
Schedule of carboplatin					
Weekly	75 (69.4)	95.4	**0.021**	0.78	1.23 (0.77–2.89)
Every 3 weeks	33 (30.6)	77.2			
PD-L1 status					
Negative	65 (60.2)	88.3	0.07		
Positive	43 (39.8)	NA			
BRCA1/2 status					
Negative	51 (47.2)	96.0	**0.009**	0.99	1.54 (0.99–3.77)
Positive	14 (13.0)	72.4			
Unknown	43 (39.8)	73.3			
Adjuvant pembrolizumab					
No	75 (69.4)	88.1	0.10		
Yes	33 (30.6)	91.1			
Adjuvant capecitabine					
No	70 (64.8)	90.2	**0.026**	0.53	1.62 (0.78–2.12)
Yes	38 (35.2)	97.8			
pCR					
Absent	39 (33.1)	87.6	**0.030**	**0.044**	1.57 (1.02–2.64)
Present	69 (63.9)	97.6			

Significant *p* values are highlighted in bold. Variables identified as significant in the univariable analysis were subsequently entered into the multivariable model. Abbreviations: ECOG—Eastern Cooperative Oncology Group; IDC—invasive ductal carcinoma; NOS—not otherwise specified; *PD-L1*—programmed death-ligand 1; *BRCA*1/2—breast cancer gene 1/2; *pCR*—pathological complete response; OS—overall survival; HR—hazard ratio; CI—confidence interval; NA—not available.

**Table 4 cancers-16-03389-t004:** Treatment-related adverse events observed in patients (n = 108) with early-stage triple-negative breast cancer.

Adverse Event	Grade 1–2, n (%)	Grade 3–4, n (%)
Alopecia	75 (69.4)	2 (1.8)
Nausea	70 (64.8)	4 (3.7)
Neutropenia	52 (48.1)	18 (16.6)
Fatigue	45 (41.6)	4 (3.7)
Vomiting	36 (33.3)	3 (2.8)
Anemia	36 (33.3)	7 (6.4)
Diarrhea	32 (29.6)	1 (0.9)
Peripheral neuropathy	26 (24.1)	2 (1.8)
Constipation	25 (23.1)	-
Increased ALT	22 (20.3)	3 (2.8)
Increased AST	21 (19.4)	3 (2.8)
Rash	21 (19.4)	-
Decreased appetite	18 (16.6)	4 (3.7)
Hypothyroidism	12 (11.1)	2 (1.8)
Stomatitis	7 (6.4)	-
Thrombocytopenia	7 (6.4)	-
Adrenal insufficiency	1 (0.9)	-

Abbreviations: ALT—alanine aminotransferase; AST—aspartate aminotransferase.

## Data Availability

Data will be available from the corresponding author upon reasonable request.

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
