# Peer review of "Neoadjuvant Pembrolizumab Plus Chemotherapy in Early-Stage Triple-Negative Breast Cancer: A Nationwide Retrospective Turkish Oncology Group Study"

_cancers, 2024, doi:10.3390/cancers16193389_

Round 1

Reviewer 1 Report

Comments and Suggestions for Authors

The present study is a retrospective analysis of oncological outcomes for early-stage TNBC patients treated with neoadjuvant pembrolizumab combined with chemotherapy across 14 centers in Türkiye. The study reports a high pathological complete response (pCR) rate of 63.9%, aligning with results from the pivotal KEYNOTE-522 trial. It also shows the strong prognostic significance of achieving pCR, which was the only independent predictor of both EFS and OS. The safety profile of pembrolizumab was consistent with clinical trial data, with most adverse events being mild to moderate. 

I have some comments before considergin potential publication:

- Lines 76-79: redundant. Remove;

- Overall, in the Introduction section, the KEYNOTE-522 very long and detailed description part should be heavily reduced or moved to the Discussion;

- The aims of your study are not clearly and concisely stated. They are too long and leave the readers very confused. Please try to condense them and try to be more effective;

- Pembrolizumab demonstrated its efficacy on oncological outcomes, but it surely allows for higher percentages of pCR (>60%) as demonstrated by the present study which subsequently influence the surgical treatment, since it converts a previously planned mastectomy into breast conservation (please cite PMID: 38539504 to improve the quality of your manuscript). The results of your study show that the majority of the analyzed patients underwent breast conserving surgery and sentinel lymph node biopsy. But, how many patients converted from a pre-planned mastectomy to breast conserving surgery?? This information is fundamental and underlines the vital importance of the implementation of pembrolizumab in TNBC. Please add this data in the Results;

- Moreover, it is certainly true that the majority of patients underwent sentinel lymph node biopsy, but how many of them were clinically node negative and how many of them were clinically/histologically node positive? This information is fundamental and should be addressed by the authors since it shows once again the utility of pembrolizumab;

- Please add data about the percentage of patients who underwent direct axillary lymph node dissection and, on the other side, the percentage of patients who underwent axillary lymph node dissection for residual disease (ypN1a/mic/itc+??) in the sentinely lymph node.

Author Response

Dear Reviewer1;  I would like to express my gratitude for taking the time to review my manuscript and for your valuable feedback and contributions.

I have made the necessary revisions in accordance with your recommendations and included the requested additional information in the manuscript.

Comment 1: - Lines 76-79: redundant. Remove;

Response 1: [Lines 76-79 were removed according to the reviewer 1’s recommendation] Thank you for pointing this out. I agree with this comment. In accordance with your suggestion, the redundant section of the article, located on the second page in the second paragraph of the introduction (lines 76-79), has been removed.

Comment 2: - Overall, in the Introduction section, the KEYNOTE-522 very long and detailed description part should be heavily reduced or moved to the Discussion;

Response 2: [In the introduction section, the part of KEYNOTE-522 trial was shortened. Afterthat, this section was revised and the part in the discussion was stated.]

-"Pembrolizumab functions by blocking the interaction between the programmed cell death protein 1 (PD-1) receptor and the PD-L1 ligand, thereby preventing the suppression of T-cell function and preserving T-cell proliferation and cytokine production [12]." This sentence, located in lines 76-79 of the introduction section, has been removed.

-"In the KEYNOTE-522 trial [11], patients with previously untreated stage II or III TNBC received either pembrolizumab or placebo combined with neoadjuvant chemotherapy, followed by surgery and adjuvant pembrolizumab or placebo. The neoadjuvant chemotherapy regimen included paclitaxel, carboplatin, and doxorubicin-cyclophosphamide or epirubicin-cyclophosphamide. The results demonstrated benefits in both neoadjuvant and adjuvant settings [11]." This sentence, located in lines 79-84 of the introduction section, has been removed. The following sentence was written in place of the one that was removed. "In the KEYNOTE-522 trial, the results demonstrated benefits in both neoadjuvant and adjuvant settings [11]." It is located in lines 80-81 of the revised article.

-"The long-term advantage of the combined neoadjuvant-adjuvant approach was evident in the improved EFS [13]. " This sentence, located in lines 86-87 of the introduction section, has been removed.

Comment3:  - The aims of your study are not clearly and concisely stated. They are too long and leave the readers very confused. Please try to condense them and try to be more effective;

Response3: [The goal of our study in the introduction was shortened and revised.]

-"This nationwide, multicenter retrospective study conducted in Türkiye sought to evaluate the real-world clinical benefits of neoadjuvant pembrolizumab combined with chemotherapy, with or without subsequent adjuvant pembrolizumab, in early-stage TNBC, reflecting physician-directed decisions on chemotherapy protocols and the variable use of adjuvant pembrolizumab with or without capecitabine." This sentence, located in lines 93-97 of the introduction section, has been removed. The following sentence was written in place of the one that was removed. "This nationwide, multicenter retrospective study conducted in Türkiye sought to evaluate the real-world clinical benefits of neoadjuvant pembrolizumab combined with chemotherapy in early-stage TNBC." It is located in lines 89-91 of the revised article.

-"We also sought to identify potential markers for treatment response and evaluate the safety profile of the implemented treatment protocols within this clinical population. By analyzing these real-life treatment patterns and outcomes, our overarching goal was to provide insights that can improve personalized care for patients with early-stage TNBC in routine clinical scenarios outside of the rigorous conditions of clinical trials." This sentence, located in lines 100-104 of the introduction section, has been removed. The following sentence was written in place of the one that was removed. "We also analyzed to identify potential markers for treatment response and evaluate the safety profile of the implemented treatment protocols within this clinical population. " It is located in lines 93-95 of the revised article

Comment 4: - Pembrolizumab demonstrated its efficacy on oncological outcomes, but it surely allows for higher percentages of pCR (>60%) as demonstrated by the present study which subsequently influence the surgical treatment, since it converts a previously planned mastectomy into breast conservation (please cite PMID: 38539504 to improve the quality of your manuscript). The results of your study show that the majority of the analyzed patients underwent breast conserving surgery and sentinel lymph node biopsy. But, how many patients converted from a pre-planned mastectomy to breast conserving surgery?? This information is fundamental and underlines the vital importance of the implementation of pembrolizumab in TNBC. Please add this data in the Results;

Response 4: "Thus, it may avoid unnecessary axillary lymph node dissection (AD) and make breast-conserving surgery (BCS) possible for patients initially considered for mastectomy. In the preoperative setting, BCS has been proven to be a safe alternative to mastectomy in terms of long-term oncologic outcomes and survival [8]." This sentence, located in lines 66-69 of the introduction, has been added to the revised version of the article.

"Reference PMID:38539504 was added as ref. No:8." Thereafter, the reference order is revised in the whole article and the reference section. (Page:13, lines:405-407)

"BCS was performed in 29 (26.8%) patients who were planned to undergo modified radical mastectomy (MRM) at the time of diagnosis after neoadjuvant treatment." This was added in the results section. It is located in lines 203-204 of the revised article.

Comment 5: - Moreover, it is certainly true that the majority of patients underwent sentinel lymph node biopsy, but how many of them were clinically node negative and how many of them were clinically/histologically node positive? This information is fundamental and should be addressed by the authors since it shows once again the utility of pembrolizumab;

Response 5: "SLNB was performed after neoadjuvant treatment in 51 of 71 patients (71.8%) with clinical/histological lymph node positivity. All of the remaining 35 patients who underwent SLNB were clinically lymph node negative."This situation was added in results section according to the suggestion of the 1 st referee. (Page:5, lines:204-207)

Comment 6: - Please add data about the percentage of patients who underwent direct axillary lymph node dissection and, on the other side, the percentage of patients who underwent axillary lymph node dissection for residual disease (ypN1a/mic/itc+??) in the sentinely lymph node.

Response 6: "Of the 22 patients who underwent axillary dissection (AD), 5 (22%) had AD without SLNB. The remaining 17 patients had AD after SLNB. Only 4 of the patients who underwent SLNB had positive N1a lymph nodes." This situation was added in the results section. ( Page:5, lines: 207-210)

Reviewer 2 Report

Comments and Suggestions for Authors

Triple-negative breast cancer remains one of the most challenging types to treat for oncologists, as mentioned by the authors.  The "perfect" regimen of chemotherapy-associated to immunotherapy remains to be precised, and the article has the merit to present the real-life results of treated patients.

The intensity of the dose could be an important issue, especially for increasing the neo antigens for overreacting lymphocytes. 

Maybe even the number of included patients may be not enough a comparison of these regimens could be interesting for readers.

Author Response

Dear Reviewer2;  I would like to express my gratitude for taking the time to review my manuscript and for your valuable feedback and contributions.

I have made the necessary revisions in accordance with your recommendations and included the requested additional information in the manuscript.

Comment 1: Triple-negative breast cancer remains one of the most challenging types to treat for oncologists, as mentioned by the authors.  The "perfect" regimen of chemotherapy-associated to immunotherapy remains to be precised, and the article has the merit to present the real-life results of treated patients.

The intensity of the dose could be an important issue, especially for increasing the neo antigens for overreacting lymphocytes. 

Maybe even the number of included patients may be not enough a comparison of these regimens could be interesting for readers.

Response 1: "No difference was found in pCR rates among the three neoadjuvant chemotherapy regimens (p:0.32)." This situation was added in the section of the response to neoadjuvant treatment. (Page: 6, lines: 230-231)

Reviewer 3 Report

Comments and Suggestions for Authors

The manuscript, entitled "Neoadjuvant Pembrolizumab Plus Chemotherapy in Early-Stage Triple-Negative Breast Cancer," The article, entitled "A Nationwide Retrospective Turkish Oncology Group (TOG) Study," reviews the results of therapy for triple-negative breast cancer using a pembrolizumab regimen in combination with chemotherapy. It is established that triple-negative breast cancer is a particularly challenging disease to treat, with a limited range of therapeutic options. The contributions of colleagues in Turkey are noteworthy and encouraging. Furthermore, the data regarding the utilization of pembrolizumab in triple-negative breast cancer is of significant practical value in the context of healthcare. This work should be published, as it represents a significant contribution to the field. As a reviewer, I am pleased to recommend that this work be published as submitted to the editorial board. In the comments section, the following observations were made: It would be beneficial to include data on five-year survival and recurrence rates. It is widely acknowledged that triple negative breast cancer frequently results in distant complications in patients.  

Author Response

Comment 1: The manuscript, entitled "Neoadjuvant Pembrolizumab Plus Chemotherapy in Early-Stage Triple-Negative Breast Cancer," The article, entitled "A Nationwide Retrospective Turkish Oncology Group (TOG) Study," reviews the results of therapy for triple-negative breast cancer using a pembrolizumab regimen in combination with chemotherapy. It is established that triple-negative breast cancer is a particularly challenging disease to treat, with a limited range of therapeutic options. The contributions of colleagues in Turkey are noteworthy and encouraging. Furthermore, the data regarding the utilization of pembrolizumab in triple-negative breast cancer is of significant practical value in the context of healthcare. This work should be published, as it represents a significant contribution to the field. As a reviewer, I am pleased to recommend that this work be published as submitted to the editorial board. In the comments section, the following observations were made: It would be beneficial to include data on five-year survival and recurrence rates. It is widely acknowledged that triple negative breast cancer frequently results in distant complications in patients.  

Response 1: Thank you very much for your valuable comments. In the current analysis of our study, we presented the EFS and OS results at a median follow-up of 19.8 months. In the future, we plan to analyze the 3- and 5-year survival results by adding new patients with longer follow-up and increasing the sample size.

Round 2

Reviewer 1 Report

Comments and Suggestions for Authors

The manuscript can be accepted in the present form.